# The Possible Pathophysiological Role of Pancreatic Stone Protein in Sepsis and Its Potential Therapeutic Implication

**DOI:** 10.3390/biomedicines12081790

**Published:** 2024-08-07

**Authors:** François Ventura, Pierre Tissières

**Affiliations:** 1Division of Anesthesiology, Geneva University Hospitals, Rue Gabrielle-Perret-Gentil 4, CH-1211 Geneva, Switzerland; 2Intensive Care Unit, Hirslanden Cliniques des Grangettes, Chemin des Grangettes 7, CH-1224 Chêne-Bougeries, Switzerland; 3Pediatric Intensive Care, Neonatal Medicine and Pediatric Emergency Department, AP-HP Paris Saclay University, Bicêtre Hospital, Le Kremlin-Bicêtre, 78 Rue du Général Lecler, 94275 Le Kremlin-Bicêtre, France; pierre.tissieres@aphp.fr; 4Institute of Integrative Biology of the Cell, CNRS, CEA, Paris Saclay University, 1 Rue de la Terrasse, 91190 Gif-sur-Yvette, France; 5Fédération Hospitalo-Universitaire FHU Sepsis, AP-HP, INSERM, Bicêtre Hospital, Paris Saclay University, 3 Rue Joliot Curie, 91190 Gif-sur-Yvette, France

**Keywords:** sepsis, pancreatic stone protein, organ failure

## Abstract

According to the current understanding of the pathophysiology of sepsis, key host dysregulated responses leading to organ failure are mediated by innate immunity, through interactions between pathogen-associated molecular patterns (PAMPs) and damaged-associated molecular patterns (DAMPs) binding to four types of pattern recognition receptors (PRRs). PRRs activation triggers the protein kinase cascade, initiating the cellular response seen during sepsis. Pancreatic stone protein (PSP), a C-type lectin protein, is a well-defined biomarker of sepsis. Studies have shown that stressed and immune-activated pancreatic β-cells secrete PSP. Animal studies have shown that PSP injection aggravates sepsis, and that the disease severity score and mortality were directly correlated with the doses of PSP injected. In humans, studies have shown that PSP activates polymorphonuclear neutrophils (PMNs) and aggravates multiple organ dysfunction syndrome. Clinical studies have shown that PSP levels are correlated with disease severity, vasopressor support, progression to organ failure, mechanical ventilation, renal replacement therapy, length of stay, and mortality. As PSP is a C-type lectin protein, it may have a role in activating innate immunity through the C-type lectin receptors (CLRs), which is one of the four PRRs. Herein, we review the literature on PSP and its possible role in the pathophysiology of sepsis, and we discuss its potential therapeutic role.

## 1. Introduction

According to the 2016 Sepsis-3 definition, “Sepsis is a dysregulated inflammatory response with organ dysfunction caused by an infection” [1]. Sepsis is a major public health problem with 48.9 million cases worldwide per year, causing 11 million deaths [2]. Sepsis and septic shock can be prevented if diagnosed and treated early using appropriate treatments, particularly a rapid administration of antibiotics. Each one-hour delay in administering antibiotics increases mortality by 8% [3]. Septic shock is defined as a subset of sepsis in which underlying circulatory and cellular metabolism abnormalities are profound enough to substantially increase mortality [1]). The latest international recommendations for the diagnosis and treatment of sepsis were published in October 2021 by the Surviving Sepsis Campaign (SSC) [4]. There is no specific treatment for sepsis and organ failure, and the current therapeutic approach includes the control of infection (with antimicrobials), the elimination of the source of infection (with drainage or surgery), and organ support of the dysregulated response of the body (organ failure), with volume resuscitation, mechanical ventilation, renal replacement therapy (RRT), and adjuvant therapies.

In children, the classical sepsis definitions were proposed in 2005, and they are like those proposed for adults in 2001 (Sepsis-2) but include the 1992 Systemic Inflammatory Response Syndrome (SIRS) criteria [5]. In January 2024, the International Consensus Criteria for Pediatric Sepsis and Septic Shock were proposed, including a new clinical and organ failure score (Phoenix Sepsis Score) [6]. The latest SSC management recommendations for sepsis in children were published in 2020 [7].

### 1.1. Role of Innate Immunity in Sepsis Pathophysiology [8]

Organ failure in sepsis is strongly associated with innate immunity activation by pathogen-associated molecular patterns (PAMPs) released by pathogen (lipopolysaccharide LPS endotoxin for Gram-negative bacteria, exotoxin, DNA sequence, others) and damaged-associated or danger-associated molecular patterns (DAMPs) released by necrotic infected cells (ATP, ADN, ARN, heat shock protein, fibrinogen, others) which bind to pattern recognition receptors (PRRs) (Figure 1).

There are four recognized PRRs: (1) Toll-like receptors (TLRs), (2) Retinoic acid inducible gene-1 like receptors (RLRs), (3) Nucleotide-binding oligomerization domain-like receptors (NLRs), and (4) C-type lectin receptors (CLRs). PRRs activation triggers the protein kinase cascade responsible for multiple cellular effects, such as the secretion of inflammatory cytokines, antimicrobial peptides, and other opsonins [8]. Once the protein kinase cascade is initiated, a variety of phases are activated (Figure 2), with complex pro- and anti-inflammatory phenomena for which a high number of research studies are currently being carried out, with the aim of identifying different phenotypes. Indeed, the immunological phenotype (hypo- vs. hyper-responsiveness) remains highly individualized and causes diagnostic and potential therapeutic difficulties. These complex pathophysiological phenomena and their therapeutic implications are regularly presented in literature reviews, for example [9].

### 1.2. Targeting Innate Immunity for Sepsis Treatment [9]

Based on the pathophysiological understanding of sepsis, therapeutic strategies can be proposed for the different phases: neutralization of PAMPs and DAMPs, activation of PRRs, activation of PRRs transduction and protein kinase, and downstream cellular effects.

With the first phase of sepsis being the binding of PAMPs and DAMPs to PRRs, the first therapeutic strategy is to prevent or rapidly reduced the release of PAMPs and DAMPs by antibiotics and control of the infectious source (surgery and drainage). Extracorporeal blood purification techniques (BPTs) have been proposed to eliminate DAMPs and PAMPs. Polymyxin B filter (a cyclic lipophilic peptide antibiotic) is known to bind Gram-negative bacteria endotoxins (or Lipopolysaccharide (LPS)), but two randomized controlled trials have shown contradictory results in terms of mortality reduction [9].

The second phase is the activation of PRRs. There are many studies on TLR agonists and inhibitor-targeted TLR activation. Unfortunately, no study ever demonstrated efficacy [10,11]. Similarly, RLR (Rig-1 like receptors) could play a role in viral infections [12]. Because of their importance in innate immunity and adjuvanticity, NLR pathways are potential targets for therapeutic strategies against auto-inflammatory disorders and sepsis [13]. Some C-type lectin receptors (CLRs) promote bacterial clearance, and they too could be targets for new prophylactic or sepsis treatment strategies [14].

The complex third phase of the protein kinase cascade activation and PRRs transduction pathways may be targeted mostly through global strategies. Among these, all therapies inhibiting or activating pathways or kinases may impact the innate immune response (e.g., adrenergic drugs, adenosine, PDE inhibitors). Finaly, downstream effectors and response may also be targeted through cytokine removal therapies (e.g., CytoSorb, oXiris) and blocking monoclonal antibodies (e.g., Anakinra, Tocilizumab). Such targeted therapies have failed to show efficacy but may be pertinent in very specific sepsis phenotypes.

Once the innate immune response has occurred, its interplay at the level of the immune synapse with adaptive immunity further complexifies the immunologic response in septic patients (Figure 2).

### 1.3. Pancreatic Stone Protein (PSP)

Pancreatic stone protein (PSP) is an early (even before symptoms = pre-symptomatic diagnostic) biomarker of sepsis used in clinical practice, and our recent comprehensive review suggests that this biomarker can be used in hospitals to screen for nosocomial sepsis in high-risk patients and to help to diagnose sepsis [15].

A 2021 meta-analysis [16] showed that the median PSP value of infected versus uninfected patients was significantly higher (81.5 ng/mL versus 19.2 ng/mL). To diagnose infection, PSP is more specific and sensitive (44.18 ng/mL AUROC 0.81–95%CI 0.78–0.85) than C-reactive protein (CRP) (99.05 mg/L—AUROC 0.77–95%CI 0.73–0.80) and Procalcitonin (PCT) (0.20 ng/mL—AUROC 0.78–95%CI 0.74–0.82).

PSP is significantly higher in patients with sepsis (146.4 ng/mL) and in patients with infections (111.4 ng/mL) compared with patients without infections (22.8 ng/mL) [17]. To diagnose sepsis, Llewelyn [18] showed on 219 patients admitted to intensive care that PSP (30 ng/mL—AUROC 0.91–95%CI 0.86–0.96) had a better accuracy than PCT (1.0 ng/mL—AUROC 0.84–95%CI 0.77–0.91). In emergency departments (n = 152), PSP (96.6 ng/mL—AUROC 0.87–95%CI 0.81–0.94) also has a better accuracy than PCT (2.02 ng/mL—AUROC 0.82– 95%CI 0.74–0.90) [16].

Several studies [19,20,21] have shown that PSP can detect infection and sepsis 3 to 5 days before the first symptoms appear, and this is the concept of the pre-symptomatic diagnosis of nosocomial sepsis. Post-cardiac surgery, PSP daily dosing (48.1 ng/mL—AUROC 0.76–95%CI 0.62–0.88) was shown to perform better than CRP (AUROC 0.53) and leucocytes (AUROC 0.64) in predicting infection [20]. In a monocentric observational study of 90 severely burned patients [19], “PSP daily measurement differentiated between sepsis, infection, and sterile inflammation up to 3 days in advance with an area under the curve of up to 0.89 (*p* < 0.001)”. An ICU prospective multicenter study [21] (14 European centers, n = 243) published in 2021 showed that following serial PSP measurements, an increase in this marker the days preceding the onset of signs and symptoms can be used to diagnose sepsis. In a prospective multicenter study of patients with complicated abdominal surgery who are at high risk of severe infections, PSP daily dosage could be an additional tool for the early detection of sepsis [22].

It is important to note that in clinical studies published before 2020, PSP values were measured using an ELISA Research Use Only (RUO) technique. Since 2020, a certified point of care nanofluidic technique (European In Vitro Diagnostic Regulation IVDR 2022, abioSCOPE^®^, Abionic, Epalinges, Switzerland) has been used for robust and rapid (<10 min) PSP measurement. The PSP clinical thresholds for infection/sepsis and the correlation between the ELISA and the nanofluidic PSP values and are detailed in our review of the literature, published in early 2024 [15].

Thirteen studies have investigated the risk and severity stratification for infection, organ failure, and sepsis. In a meta-analysis of five studies [18,23,24,25,26] (n = 678) [27], PSP and PCT (respectively, AUROC 0.80–95%CI 0.75–0.85 and AUROC 0.79–95%CI 0.74–0.84) had a better performance than CRP (AUROC 0.56–95%CI 0.50–0.63) in discriminating mild infection from severe infection, sepsis, and septic shock. In predicting the 28-day mortality rate, PSP (AUROC 0.69–95%CI 0.64–0.74) was more accurate than PCT (AUROC 0.61–95%CI 0.56–0.66) and CRP (AUROC 0.52–95%CI 0.47–0.57). PSP was associated with the SOFA score in a study on ventilator-associated pneumonia (VAP) (n = 101) from VAP onset (Spearman rank correlation coefficient 0.49, *p* < 0.001) up to day 7 [28]. In another study with 107 ICU patients, the risk of 28-day mortality rate increased continuously for each ascending quartile of PSP level at admission [29]. In a study with 141 ICU patients [30], PSP was correlated with mechanical ventilation (r  =  0.607; *p*  <  0.01), vasopressor administration at admission and during the stay (respectively, r  =  0.496; *p*  <  0.001 and r  =  0.545; *p*  <  0.001), SOFA score (*p* < 0.001), and renal replacement therapy (RRT) (r  =  0.360; *p*  =  0.015).

All of these clinical studies suggest a potentially central role of PSP in sepsis pathophysiology. The aim of this article is to present the existing data on PSP and review its role in the pathophysiology of sepsis.

## 2. Review Objectives and Methods

The objectives are to review physiological and the pathophysiological PSP studies to develop hypotheses on the potential role of PSP in the diagnosis and treatment of sepsis.

The literature search was performed in April 2024 on PubMed databases using pancreatic stone protein, PSP, PSP/reg, regenerating protein (REG1), lithostathine, infection, and sepsis, as keywords and/or MeSH Terms.

## 3. Results

### 3.1. Definition and Origin of Pancreatic Stone Protein (PSP)

As explained in our recent literature review [15], in over 600 publications since the 1970s, PSP has been described as being secreted mainly by the pancreas. Some teams have conducted research on the exocrine function of the pancreas and others on the endocrine function [31].

Firstly, a protein secreted by the acinar cells of the pancreas, then called lithostathine, was discovered during research into pancreatitis [32]. Lithostathine was hypothesized to inhibit calcium carbonate crystal precipitation in the pancreatic juice and prevent the formation of pancreatic stones. Even if this function has proved to be false, the name lithostathine has been replaced by pancreatic stone protein [33,34].

Secondly, an islet-derived protein secreted by β-cells of the islets of Langerhans was discovered during research into diabetes [35]. This protein has potential β-cell regenerative activity, and it has been classified in the regenerative Reg or REG protein family, and named REG1.

It subsequently emerged that PSP (formerly lithostathine) and REG1 were very similar [36]. The name PSP is used today to describe both the PSP and the REG1, although the names PSP/REG1 or PSP/reg would be more accurate. It should also be noted that pancreatic acinar cells play an important role in the development and maintenance of islets of Langerhans β-cells [37].

An immunohistochemical analysis of healthy patients showed that PSP is predominantly found in the pancreas, followed in order of importance by the duodenum, jejunum, ileum, blood, stomach, colon, kidney, and liver [38].

### 3.2. Physiology of Pancreatic Stone Protein (PSP)

The PSP secreted in the duodenum by acinar cells is a 16 KDa 144-amino-acid glycoprotein. In pancreatic juice, trypsin digests this 16 KDa PSP into a 14 KDa PSP (133-amino-acid glycoprotein—lacking an 11-amino-acid N-terminal peptide) [37]. In the digestive tract, this 14 KDa PSP is insoluble and cannot be reabsorbed in the blood. PSP measured in the blood does not therefore reflect the 14 KDa PSP level. In human cadavers, immunohistochemical studies have shown that it is not the 14 KDa PSP that increases during sepsis but the 16 KDa PSP [38]. It could be that the 16 KDa PSP is released by damaged acinar cells or the 16 KDa PSP/REG1 secreted by pancreatic β-cells [39] (Figure 3).

The physiological function of the 14 KDa PSP has not been fully elucidated [40]. For unknown reasons, the blood level of PSP increases in 79% of cases of acute pancreatitis, 44% of chronic pancreatitis, 42% of pancreatic cancer, 33% of gastric cancer, 19% of liver cirrhosis, 18% of gallstone, and 11% of peptic ulcer [41].

The PSP level is always increased in cases of chronic renal failure, and 17% in diabetes mellitus [40]. The origin of this increase, in the absence of pancreatic lesions, was likely related to PSP renal clearance impairment and diabetic β-cell activation.

PSP is structurally similar to C-type lectin-like proteins [37,42], which are calcium-dependent glycan-binding proteins involved in the process of cell-to-cell and host–cell interactions, including adhesion and signaling receptors in homeostasis and innate immunity, as well as being involved in leukocyte and platelet trafficking during inflammation [42].

### 3.3. Pathophysiology of Pancreatic Stone Protein (PSP) in Infection and Sepsis [15]

The 14 KDa PSP secreted in the duodenum by acinar cells binds to bacteria [43] (via aggregation and immobilization), which could partly explain the antibacterial role of pancreatic juice.

In 2002, it was discovered by chance that 16 KDa PSP blood level in animals increased during anesthesia stress. In 2009, in a study of 83 trauma patients [17], PSP blood levels doubled after trauma (22.8 ng/mL vs. healthy controls 10.4 ng/mL) but increased by more than 10 times in trauma patients with infection (111.4 g/mL) and sepsis (146.4 ng/mL). Since the pancreas was not traumatized (normal CT-scan, amylase, and lipase plasma levels), it was hypothesized that the 16 KDa PSP plasma increase was not explained by damaged acinar cells release, but through an unclear mechanism (or eventually by enterocytes).

The exact function of PSP in infection and sepsis remains unknown. In humans, PSP binds to polymorphonuclear neutrophils (PMNs) and activates PMNs and microcirculatory failure by decreasing CD62L and increasing CD11b [17]. In mice injected with phosphate-buffered saline (PBS), low-dose (systemic infection dose) or high-dose PSP (same concentration seen during sepsis) following cecal ligation and puncture (CLP), the level of inflammatory factors (TNF-α, IL-6, IL1-β), key parameters reflecting organ failure (lactate dehydrogenase, troponin, creatinine, lung wet/dry ratio), the disease severity score, and mortality were directly correlated with the doses of PSP injected [30]. Immunofluorescence staining revealed that after PSP administration, PMN tissue infiltration was present in a dose-dependent manner in several important organs (kidney, hearts, lungs, and liver) suggesting that PSP plays a role in enhancing PMN infiltration (confirmed by the increased levels of intracellular adhesion molecules ICAM-1 and CD29), which leads to aggravated multiple organ dysfunction syndrome (MODS). Table 1 summarizes studies showing the correlation between PSP and sepsis.

## 4. Discussion

Following our review, the theory most frequently cited since 2009 in the literature to explain the increase in 16 KDa PSP during infection and sepsis, namely that of secretion by pancreatic acinar cells, does not seem plausible. The hypothesis could therefore be that it is an increase in the 16 KDa PSP (or REG1) secreted by pancreatic β-cells that is observed in the blood measurements during infection and sepsis.

The secretion of 16 KDa PSP by β-cells could be induced by immune activation following infection, and/or a dysregulated immune reaction in the context of sepsis with pro- and anti-inflammatory cytokines release [17,44]. The links between immune activation and β-cell responses [45], between sepsis and insulin [46], and between β-cell stress and PSP [47] are described.

PSP is generally described as an inflammatory acute-phase protein (APP) [17], but this theory is wrong. PSP is produced mainly by the pancreas, and then APPs are by definition produced by the liver, and PSP does not increase during inflammation [19,20,48,49].

Although studies are still needed to elucidate the pathways of sepsis and PMN activation, it can be hypothesized that during sepsis, PSP (a C-type lectin-like protein) might be a DAMP-like activating C-type lectin receptors (CLRs) (one of the four PRRs; see Figure 1) involved in the host response to pathogens by activating PMNs, thus reflecting the protein kinase cascade activation and progression to MODS. As mentioned, in mice, PSP injection aggravates sepsis [30]. In humans, the PSP level is correlated with disease severity, SOFA score, vasopressor support, RRT, progression to organ failure, mechanical ventilation, treatment escalation, length of stay, and mortality [27,28,29,30] (Table 1).

Based on this literature review, we propose a new original hypothetic model of the role of PSP in sepsis in which PSP, secreted by pancreatic β-cells [15] upon the stimulation of these cells during infection and sepsis [47], is linked to innate immune activation through CLR binding (Figure 4).

This new paradigm may suggest that PSP could contribute to organ failure, sepsis, and septic shock and that PSP modulation could impact sepsis pathophysiology. Accordingly, innate immunity immunomodulation through targeted therapy on PSP should be investigated.

## 5. Conclusions

In this study, we reviewed roles and functions of PSP in the innate immune response to infection. PSP measured in blood is that secreted by β-cells is probably not carried out by pancreatic acinar cells. PSP is not an acute phase protein and could be considered as a DAMP-like activating C-type lectin receptors (CLRs), therefore positioning PSP as a potent regulator of the innate immune response.

This new understanding of the possible role of PSP and its potential involvement as an early mediator of sepsis and organ failure could lead to a novel avenue in treating sepsis.

Animal and human studies are needed to corroborate these hypotheses.

## Figures and Tables

**Figure 1 biomedicines-12-01790-f001:**
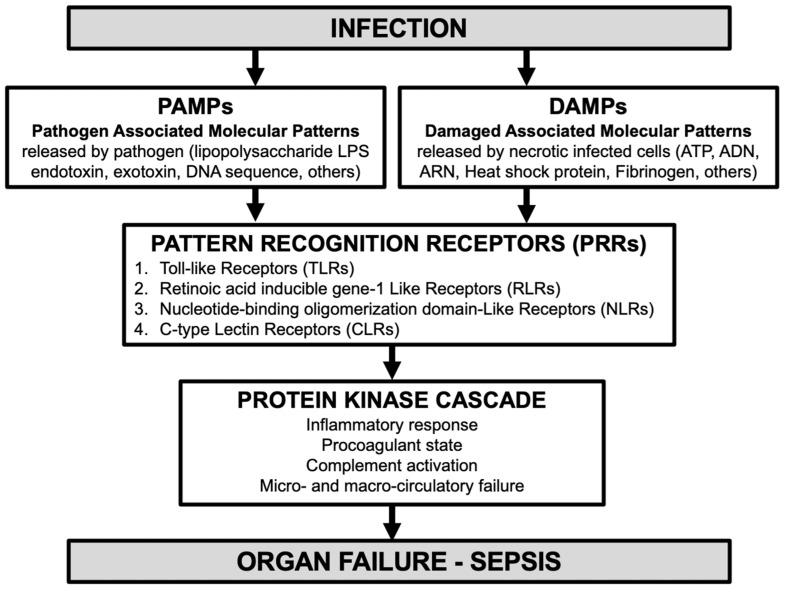
Physiopathology of organ failure in the case of sepsis (adapted [8]). Pathogen-associated molecular patterns (PAMPs) released by pathogen and damaged-associated or danger-associated molecular patterns (DAMPs) released by necrotic infected cells bind to 4 different pattern recognition receptors (PRRs). These activations trigger the protein kinase cascade, leading to organ failure and sepsis.

**Figure 2 biomedicines-12-01790-f002:**
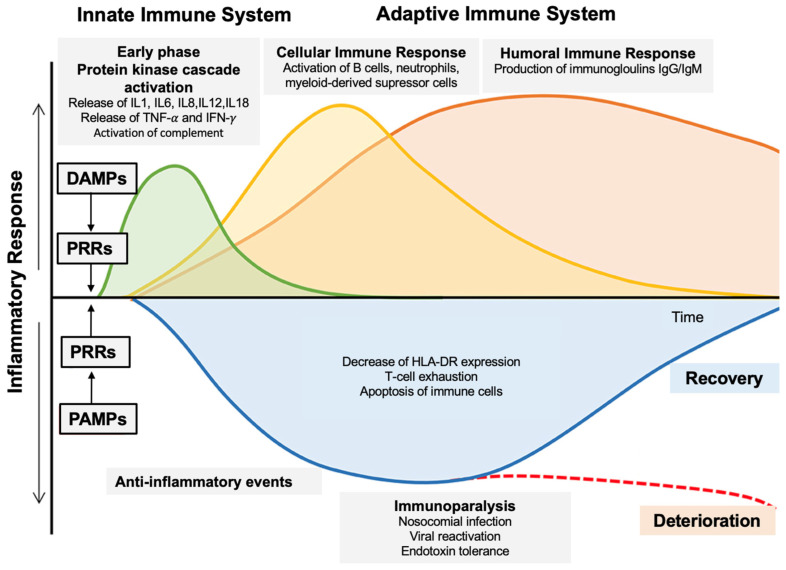
Complex pathophysiology of sepsis concepts (adapted from [9]). Globally, sepsis is initiated by the activation of the protein kinase cascade (Figure 1) with an early immune response followed by pro-/anti-inflammatory responses with cellular and humoral immune responses, then a phase of immunoparalysis before a phase of recovery or deterioration. Abbreviations: DAMPs, damaged-associated or danger-associated molecular patterns; HLA-DR, human leukocyte antigen-D related; IgM/G, immunoglobulin M/G; IL, interleukin; IFN-y, Interferon y; PAMPs, pathogen-associated molecular patterns; PRRs, Pattern recognition receptors; TNF-α, tumor necrosis factor alpha.

**Figure 3 biomedicines-12-01790-f003:**
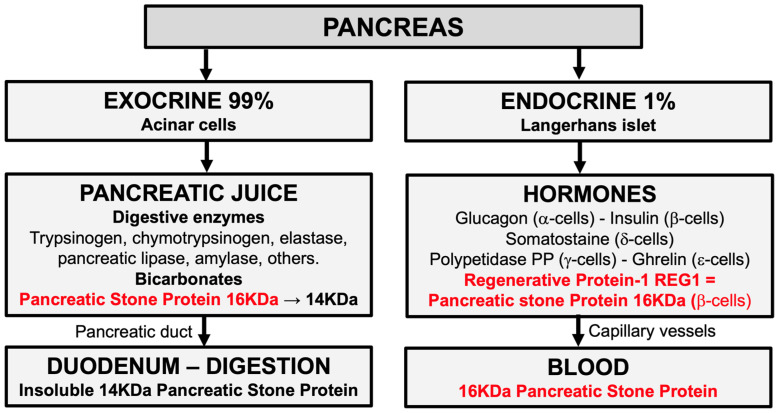
The endocrine 16 KDa pancreatic stone protein (PSP) and the insoluble 14 KDa exocrine/digestive PSP.

**Figure 4 biomedicines-12-01790-f004:**
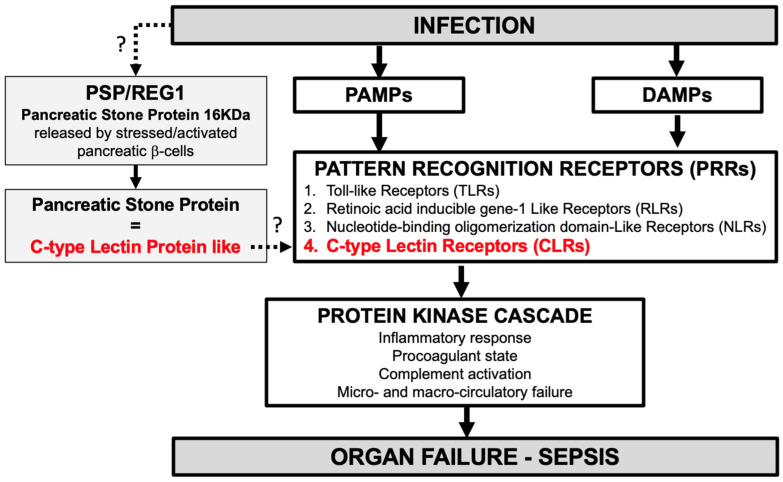
Theoretical new hypothesis model of the pathophysiology of pancreatic stone protein (PSP) (=Regenerative Protein1 (REG1)) in sepsis.

**Table 1 biomedicines-12-01790-t001:** Pancreatic stone protein (PSP) and sepsis correlation.

M. Keel et al., 2009 [17] Human clinical study	PSP activates polymorphonuclear neutrophils and microcirculatory failure.
CX. Jin et al., 2011 [37] Literature review	PSP is structurally similar to C-type lectin-like proteins.
L. Boeck et al., 2011 [28] Clinical study	PSP was correlated with the SOFA score from ventilator-associated pneumonia (VAP) onset up to day 7.
P. Hu et al., 2023 [30] Animal study	After PSP injection in mice, level of inflammatory factors, PMN tissue infiltration, key parameters reflecting organ failure, disease severity score, and mortality were directly correlated with the doses of PSP injected.
P. Hu et al., 2023 [30] Clinical study	PSP was correlated with mechanical ventilation, vasopressors administration at admission and during the stay, SOFA score, and renal replacement therapy (RRT).
P. Zuercher et al., 2023 [27] Meta-analysis	PSP discriminates mild infection, severe infection, sepsis, and septic shock. The 28-day mortality rate increased continuously for each ascending quartile of PSP level at admission.

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
