# Peer review of "The Possible Pathophysiological Role of Pancreatic Stone Protein in Sepsis and Its Potential Therapeutic Implication"

_biomedicines, 2024, doi:10.3390/biomedicines12081790_

Round 1

Reviewer 1 Report

Comments and Suggestions for Authors

I am glad that the authors accumulated and described existing data on “Pancreatic Stone Protein and review its role in the pathophysiology of sepsis”. The findings are novel with well scientific explanation. However, there is a minor comment before publishing it.

All the figure’s quality are poor with hard to read. So, the image needs to be improved with bigger font size and high picture quality

Author Response

Comments1: All the figure’s quality are poor with hard to read. So, the image needs to be improved with bigger font size and high picture quality

Response1: We've redesigned the figures with larger font size and colour to improve visibility. The images transmitted are of the highest quality.

Thanks,

Best regards

Reviewer 2 Report

Comments and Suggestions for Authors

In the present manuscript, Ventura and Pierre have conducted a literature review on PSP, examining its role in the pathophysiology of sepsis and discussing its potential therapeutic applications. This compilation of literature is commendable; however, some modifications are necessary to achieve an acceptable form. Please refer to the comments provided below:

1. Expand the abbreviation used in their first use. e.g., SSC 

2. The authors tried to hypothesize the plausible role of PSP in sepsis and tried to see the therapeutic implications. However, the correlation was established majorly with the inflammation and with pancreatic cells and their secretion. The review lacks the evidences in establishing the linkage between the PSP with stress and its diagnostic and possible management. Thus, the authors are suggested to retrieve and review study suggesting this relation clinically and summarize the information in tabular form.

3. The methodology of the review is not clear, as author fails to explain the retrieval scheme, detailed search strategy, selection of articles, minimizing possible conflicts and inclusion of the actual article. Authors are suggested to prepare the chart on procedure followed during the conduct of review.

4. In Figure 3, the 16KDa PSP endocrine form and the 14KDa exocrine PSP form depicted in the physiology section does not shows the physiological role of PSP in body rather just explain the formation of same. Make the figure more comprehensive by adding its physiological actions and if possible depict its role in inflammation and sepsis.

5. New hypothesis is raised in the discussion section. However, we suggest authors to establish the gross relation between the PSP and sepsis in the result section.  

6. Section 1.3, 3.1, 3.2, and entire discussion section are heavily plagiarized showing plagiarism of Approximate 40%. Try to keep the similarity index within Journal standard.

Comments on the Quality of English Language

Minor punctuation errors are there those can be omitted while proofreading.

Author Response

Comments1. Expand the abbreviation used in their first use. e.g., SSC

Response1: done

Comments2. The authors tried to hypothesize the plausible role of PSP in sepsis and tried to see the therapeutic implications. However, the correlation was established majorly with the inflammation and with pancreatic cells and their secretion. The review lacks the evidences in establishing the linkage between the PSP with stress and its diagnostic and possible management. Thus, the authors are suggested to retrieve and review study suggesting this relation clinically and summarize the information in tabular form.

Response2: In the article we assume a possible link between infection, beta cell stimulation and PSP secretion (line 243-247). 

"The secretion of 16 KDa PSP by b-cells could be induced by immune activation following infection, and/or a dysregulated immune reaction in the context of sepsis with pro- and anti-inflammatory cytokines release [17] [45]. The link between immune activation and b-cell responses [44], between sepsis and insulin [46], and between b-cell stress and PSP [47] are known".

With regard to the function of PSP during sepsis, we cite some fundamental studies (line 251-263). 

"In human, PSP binds to polymorphonuclear neutrophils (PMNs) and activates PMNs and microcirculatory failure by decreasing CD62L and increasing CD11b [17]. In mice injected with phosphate buffered saline (PBS), low- (systemic infection dose), and high-dose PSP (same concentration seen during sepsis) following cecal ligations and puncture (CLP), level of inflammatory factors (TNF-a, IL-6, IL1-b), key parameters reflecting of organ failure (Lactate dehydrogenase, troponin, creatinine, lung wet/dry ratio), disease severity score and mortality were directly correlated with the doses of PSP injected [30]. Immunofluorescence staining revealed that, after PSP administration, PMN tissue infiltration was present in a dose dependent manner in several important organs (kidney, hearts, lungs, and liver) suggesting that PSP plays a role in enhancing PMN infiltration (confirmed by the increased levels of intracellular adhesion molecule ICAM-1 and CD29), which leads to aggravated multiple organ dysfunction syndrome (MODS)".

The value of PSP correlates with the severity of sepsis in human clinical studies (line 148-161) and line .

"Thirteen studies investigated risk and severity stratification for infection, organ failure and sepsis. In a meta-analysis of 5 studies [18] [23] [24] [25] [26] (n=678) [27] PSP and PCT (respectively AUROC 0.80 - 95%CI 0.75–0.85 and AUROC 0.79 - 95%CI 0.74–0.84) had a better performance than CRP (AUROC 0.56 - 95%CI 0.50–0.63) to discriminate mild infection and severe infection, sepsis and septic shock. To predict 28-day mortality rate,  PSP (AUROC 0.69 - 95%CI 0.64-0.74) was more accurate than PCT (AUROC 0.61 - 95%CI 0.56-0.66) and CRP (AUROC 0.52 - 95%CI 0.47-0.57). PSP was associated with the SOFA score in a study on ventilator associated pneumonia (VAP) (n=101) a from VAP onset (Spearman rank correlation coefficient 0.49, p< .001) up to day 7 [28]. In a 107 ICU patients study risk of 28-day mortality rate increased continuously for each ascending quartile of PSP level at admission [29]. In a 141 ICU patients study [30], PSP was correlated with mechanical ventilation (r = 0.607; p < 0.01), vasopressors administration at admission and during the stay (respectively r = 0.496; p < 0.001 and r = 0.545; p < 0.001), SOFA score (p<0.001) and renal replacement therapy (RRT) (r = 0.360; p = 0.015)".

"As mentionned, in mice, PSP injection aggravates sepsis [30]. In human, PSP level is correlated with disease severity, SOFA score, vasopressor support, RRT, progression to organ failure, mechanical ventilation, treatment escalation, length of stay, mortality [27] [28] [29] [30]". 

To sum up, we think we've already answered your comment No. 2 in the article, but we've added a table as suggested. 

Comments3. The methodology of the review is not clear, as author fails to explain the retrieval scheme, detailed search strategy, selection of articles, minimizing possible conflicts and inclusion of the actual article. Authors are suggested to prepare the chart on procedure followed during the conduct of review.

Response3. Our article is a review of the literature, not a meta-analysis. At the time of submissionn, the editors even asked us not to talk about the method. So, at their request, we explained the minimum. In our review of the clinical literature published at the beginning of 2024 (Ventura et al. Medical Research Archives https://doi.org/10.18103/mra.v11i12.4893) we go into more detail about the 52 clinical studies on PSP and sepsis. 

Comments4. In Figure 3, the 16KDa PSP endocrine form and the 14KDa exocrine PSP form depicted in the physiology section does not shows the physiological role of PSP in body rather just explain the formation of same. Make the figure more comprehensive by adding its physiological actions and if possible depict its role in inflammation and sepsis.

Response4.At the request of another reviewer, we have already modified figure 3 to make it more visible. The aim of this figure is to clearly show the difference between the 2 PSPs, not to talk about their physiological function, which is still unclear. 

Comments5. New hypothesis is raised in the discussion section. However, we suggest authors to establish the gross relation between the PSP and sepsis in the result section.  

Response5. Good point and we've made the change. 

Comments6. Section 1.3, 3.1, 3.2, and entire discussion section are heavily plagiarized showing plagiarism of Approximate 40%. Try to keep the similarity index within Journal standard.

Response6: We are surprised by your analysis. The publishers/journal have already tested our article with their plagiarism detection programme. It is true that we cite articles and that we cite our own review of the clinical literature (reference 15 - Ventura et al 2023, Medical Research archives https://doi.org/10.18103/mra.v11i12.4893). We have therefore already made changes to cite the articles we refer to differently, which has enabled us to achieve 0% plagiarism with MDPI programme, and thus meet the journal's quality criteria.

Our article and especially our hypotheses on the potential role of PSP are completely new and there is no plagiarism.  On the contrary, the theories (PSP is an acute phase protein secreted by acing cells) repeated (without analysis) in all the introductions to articles on PSP since 2009 are not reproduced in our article. The existing studies used to establish our hypotheses are all referenced in the article. 

Thank you for all your comments and we hope we have been able to respond to them.

Best regards

Reviewer 3 Report

Comments and Suggestions for Authors

Manuscript titled "The Pathophysiological Role of Pancreatic Stone Protein (PSP) in Sepsis and its Possible Therapeutic Implication". An interesting knowledge has been proposed. The manuscript does an excellent job of comprehensively reviewing the existing literature on Pancreatic Stone Protein (PSP) and its role in sepsis. By collating findings from numerous studies, it provides a clear and well-supported overview of PSP's potential as both a biomarker and a therapeutic target. This thorough synthesis of current research is invaluable for advancing our understanding of PSP and guiding future investigations. However the following comments should be addressed before acceptance

Comments

Introduction Clarification: The introduction provides a general overview of sepsis. Can you include a brief section on the discovery and historical context of PSP to set the stage for its role in sepsis

Figures Explanation: Figures 1 and 2 illustrate the pathophysiology of sepsis and the complex interactions involved. Suggested to provide a more detailed explanation or legend for these figures to help readers better understand the depicted processes

The manuscript mentions that PSP levels are significantly higher in sepsis patients compared to non-sepsis patients. Authors should provide the specific numerical ranges for PSP levels across different studies to illustrate this difference more clearly

The manuscript posits that PSP plays a significant role in sepsis pathophysiology, but the exact mechanistic pathways remain unclear. Author should discuss more rigorous experimental data or propose specific molecular mechanisms by which PSP influences immune responses and organ dysfunction in sepsis? Without this mechanistic clarity, the causal relationship between PSP levels and sepsis outcomes remains speculative.

Author Response

Comments1: Introduction Clarification: The introduction provides a general overview of sepsis. Can you include a brief section on the discovery and historical context of PSP to set the stage for its role in sepsis.

Response 1: In the introduction we discuss the clinical utility (discovered by chance) of PSP in the diagnosis of sepsis. The aim of our literature review is to gain a better understanding of the physiology and pathophysiology of PSP in sepsis and thus try to understand why PSP increases in sepsis. Indeed, its role is not very clear. You will therefore find the background to the physiology of PSP in the results (3.1 Definition and origine of PSP - 3.2. Physiology - 3.3. Pathophysiology) of our review and not in the introduction. 

Comments2: Figures Explanation: Figures 1 and 2 illustrate the pathophysiology of sepsis and the complex interactions involved. Suggested to provide a more detailed explanation or legend for these figures to help readers better understand the depicted processes. 

Response 2: Our article will be included in a special edition of the journal Biomedicine devoted solely to sepsis (Sepsis and Septic Shock: From Molecular Mechanism to Novel Therapies). In order to avoid all article introductions saying the same thing every time, and assuming that entire articles will be devoted to the pathophsyiology of sepsis, we have deliberately not given all the details in order to focus on our hypothesis. This is by choice. But we have completed the figure legends as desired. 

Comments3: The manuscript mentions that PSP levels are significantly higher in sepsis patients compared to non-sepsis patients. Authors should provide the specific numerical ranges for PSP levels across different studies to illustrate this difference more clearly.

Responses 3:  Before the 2020s, PSP was measured using a ELISA technique (Research Use Only RUO). Since 2020, PSP can be measured using a certified nanofluidic technique (European In Vitro Diagnostic Regulation 2022 - IVDR 2022). A correlation between the PSP levels measured by the RUO ELISA and the nanofluidique can be used to compare PSP studies before and after 2020 (nanofluidique ng/ml = 4.6 x RUO ELISA ng/ml + 30 ng/ml). PSP values must therefore be interpreted according to the assay technique used (differences being explained by the type of antibody used and the sensitivity of the technology). All this is explained in detail in our review of the literature on PSP clinical studies published in early 2024. (reference 15 - Ventura et al. Medical Research Archives https://doi.org/10.18103/mra.v11i12.4893 ). In order not to complicate our article on physiology and pathophysiology, we have decided not to add these explanations on the interpretation of clinical PSP values. We have therefore added these explanations to the text of the article:

"For information, in clinical studies published before 2020, PSP values were measured using an ELISA-Research Use Only (RUO) technique. Since 2020, a certified nanofluidic technique (European In Vitro Diagnostic Regulation IVDR 2022, abioSCOPE, Abionic, Epalinges, Switzerland) has been used for robust PSP measurement. The PSP clinical thresholds for infection/sepsis, and the correlation between the ELISA and the nanofluidic PSP values and are detailed in our review of the literature published in early 2024 [15]". 

Comments4: The manuscript posits that PSP plays a significant role in sepsis pathophysiology, but the exact mechanistic pathways remain unclear. Author should discuss more rigorous experimental data or propose specific molecular mechanisms by which PSP influences immune responses and organ dysfunction in sepsis? Without this mechanistic clarity, the causal relationship between PSP levels and sepsis outcomes remains speculative.

Responses4: Indeed, we formulate a hypothesis based on current data suggesting that PSP is more a mediator of sepsis than just a marker. The PSP exact mechanism is not known and we propose to carry out studies to understand it better. In our discussion and conclusion, we make it clear that this is a hypothesis.

"Based on this literature review, we proposed a new original theoretical hypothetic model of the role of PSP in sepsis in which PSP, secreted by pancreatic b-cells [15] upon stimulation of these cells during infection and sepsis [47], is linked to innate immune activation through CLR binding (Figure 4)". 

But, we've changed the title of the article to make it clearer that these are hypotheses. The Possible Pathophysiological Role of Pancreatic Stone Protein in Sepsis and its Potential Therapeutic Implication. The abstract and conclusion have also been changed slightly to make it clear that this is a hypothesis. Prof Tissières plans to carry out fundamental research studies in his laboratory.

Thank you for all your comments and we hope we have been able to respond to them.

Best regards